biological applications, ecology, behaviour

marine protected areas (MPAs), selective harvesting, density-dependence, European lobster, phenotypic rescue, marine reserves

**Author for correspondence:**
Tonje Knutsen Sørdalen
e-mail: tonjesordalen@gmail.com

†These authors contributed equally to this work.

# Protection from fishing improves body growth of an exploited species

Tonje Knutsen Sørdalen[1,2,†], Kim Tallaksen Halvorsen[2,†] and Esben Moland Olsen[1,2]

[1]Centre for Coastal Research, Department of Natural Sciences, University of Agder, N-4604 Kristiansand, Norway
[2]Institute of Marine Research, Flødevigen, Nye Flødevigvei 20, N-4817 His, Norway

TKS, 0000-0001-5836-9327; KTH, 0000-0001-6857-2492; EMO, 0000-0003-3807-7524

Hunting and fishing are often size-selective, which favours slow body growth. In addition, fast growth rate has been shown to be positively correlated with behavioural traits that increase encounter rates and catchability in passive fishing gears such as baited traps. This harvest-induced selection should be effectively eliminated in no-take marine-protected areas (MPAs) unless strong density dependence results in reduced growth rates. We compared body growth of European lobster (*Homarus gammarus*) between three MPAs and three fished areas. After 14 years of protection from intensive, size-selective lobster fisheries, the densities in MPAs have increased considerably, and we demonstrate that females moult more frequently and grow more during each moult in the MPAs. A similar, but weaker pattern was evident for males. This study suggests that MPAs can shield a wild population from slow-growth selection, which can explain the rapid recovery of size structure following implementation. If slow-growth selection is a widespread phenomenon in fisheries, the effectiveness of MPAs as a management tool can be higher than currently anticipated.

## 1. Introduction

Intense human harvest of wild animals has caused reductions in body size and eroded age structure in exploited populations worldwide [1,2]. Such downsizing can be a result of increased mortality alone but is often reinforced by selective harvesting practices. For example, many fisheries must avoid catching juveniles and small adults, either through regulations applied to the fishing gear (hook size, mesh size, trap entrance size), or through minimum size limits [3]. Similarly, rules that are intended to prevent the killing of small and young individuals are commonly applied in management of hunted mammal populations [4]. In addition to size selectivity imposed for management or conservation reasons, the largest animals have often a disproportional economic or experiential value to hunters and fishers. Fishing also has the potential to selectively remove fast-growing individuals if growth rate is positively correlated with behaviour traits that increase vulnerability of capture [5,6].

Harvest-induced downsizing is typically assumed to have impacts, mostly negative, on population dynamics and ecological interactions [2]. This is because body size tends be positively correlated with fitness-related traits in both males and females. For example, big parents often produce more viable offspring, providing them with more nutritional resources and/or better care and protection [7,–9]. In fish and crustaceans, animals with indeterminate growth, large females of most species produce disproportionally more offspring relative to smaller females [10], while large males tend to be preferred mating partners [11–13]. For species with sexual-size dimorphism, size-selective harvesting can also lead to skewed sex ratios, disrupted mating behaviour and cause gamete-limitation [13–15]. Selective harvesting can also cause ecosystem-level consequences, for instance, downsizing of key predatory species can reduce their effectiveness in controlling prey population and result in negative ecological cascades [16,17] (but see [18]). Further, populations that have undergone size and age truncation can have reduced resilience to cope with environmental change [19–23].

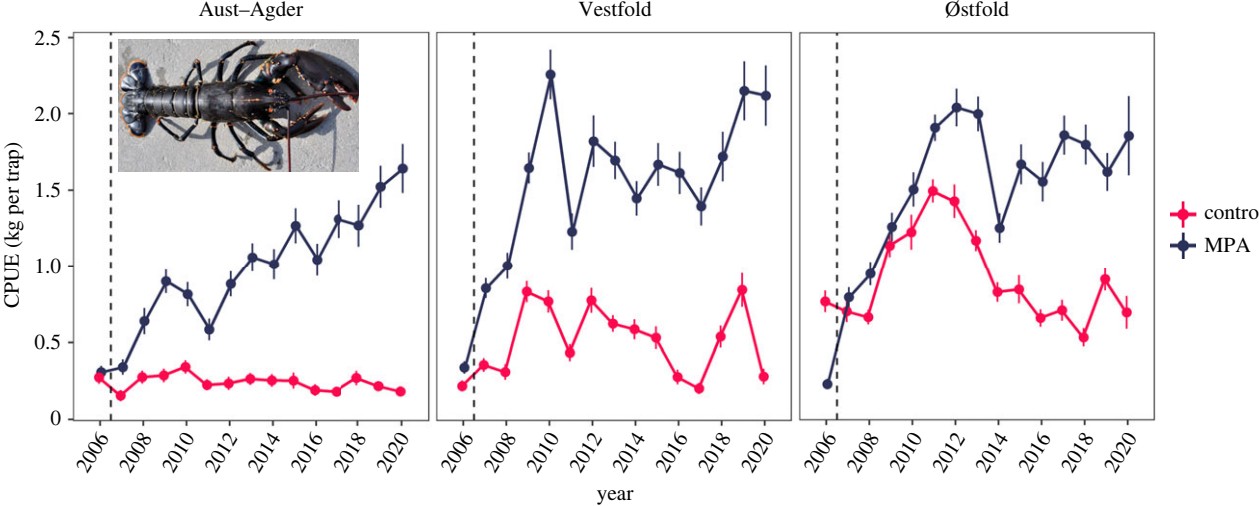

**Figure 1.** The development in mean CPUE of all European lobster caught in the annual research trap survey in lobster reserves and fished areas of (*a*) Aust–Agder, (*b*) Vestfold and (*c*) Østfold between 2006 and 2020 (modified from Knutsen *et al.* [42]). Establishment of the protected areas from August of 2006 is indicated by vertical dashed line. The error bars depict s.e. around the mean. For weight–length relationship, see model in electronic supplementary material, figure S3. (Online version in colour.)

An effective way of mitigating consequences of selective harvesting is to establish no-take protected areas, reserves where the removal of all or specific species is prohibited. The premise is simple; if large enough and well-enforced, protected areas provides opportunity for species to restore abundances, size and age structure and other types of phenotypic complexity [24–26]. In addition, neighbouring un-protected areas can benefit through spill-over of surplus recruits and unselected adults [27,28]. Protected areas, and un-protected control areas with similar ecological characteristics, provide unique opportunities to study how harvesting affects trait distributions and population dynamics [29]. This is a necessary study design for disentangling the effects of harvest selection and environmental factors on phenotypic traits [4]. In the marine environment, there is now ample evidence that marine-protected areas (MPAs) can improve body size and other morphological traits [24,30–32], but less is known about how life-history traits, such as somatic growth rate, is affected [33,34]. Is the increased body size in MPAs solely due to the upheaval of fishing mortality, or is it also modulated by shifts in somatic growth? If an MPA does improve body growth (due to elimination of harvest selection against fast growth rate), this should have additional positive effects on population productivity and contribute to an even faster restoration of size structure. Alternatively, increased population densities in protected areas may intensify competition and cause reduced body growth rates, which would constrain recovery and reduce the benefits off spill-over to harvested areas [34–36].

In this study, we provide a robust empirical assessment of how body growth responds to protection from selective fishing mortality in a heavily exploited population of European lobster. We analysed individual body growth using 14 years of capture–recapture data from three pairs of no-take lobster MPAs and adjacent areas open to fishing in Skagerrak, southern Norway. In this region, lobsters are subjected to an intensive trap fishery which has led to gradual and strong reductions in catch rates across several decades [37]. We regard it as likely that the lobster fishery is targeting fast-growing individuals for three reasons: first, the lobster fishery is size-selective [38], which should select for slower growth [39]. Lobsters grow by replacing their exoskeleton (moulting). This implies lower

fishing mortality for slow growers; more specifically for the individuals (smaller than 250 mm) that either skip moulting or that do not surpass the size limit if they moult. Second, female lobsters that carry visible eggs must be returned to sea. Females do not moult when they are berried, and they make up approximately 50% of the female catch during the fishing season [40]. This regulation may therefore induce additional selective pressure for skipped moulting in females, favouring those that invest surplus energy in eggs rather than somatic growth. Third, it is also probable that fast-growing lobsters have higher probability of being captured in the baited traps, independently of body size. This in line with the hypothesis on behaviour-driven growth selectivity in passive gears, and especially since the most convincing experimental evidence is from a species of clawed crayfish (*Cherax destructor*) [6], which have many morphological and behavioural similarities with clawed lobsters. In that study, individual boldness and voracity was found to correlate with fast growth rate, and as bold and fast-growing individuals spent more time searching for food, they also were more likely to encounter and enter baited traps. Unfortunately, similar experiments have not been conducted on clawed lobsters; however, a recent study finds support for behaviour-driven selection in lobster in Southern Norway, where males with large claws relative to body size has higher capture probability in the fishery, and consequently males in fished areas have smaller claws compared to males inside MPAs [32,41]. Lobsters in the MPAs are protected from these selective pressures and are therefore predicted to grow relatively faster—a difference that should increase with body size due to the size-selective fishing regulations and the cumulative effects of assumed growth selection over several seasons. Alternatively, it is possible that density dependence is strong and have resulted in lower growth rates for lobsters in MPAs where the density of lobsters has increased substantially in MPAs since implementation (figure 1).

## 2. Methods and materials

### (a) Study species and study system

European lobsters are large and long-lived crustaceans exhibiting sexual dimorphism. Males grow faster than females,

mature at smaller size and have relatively larger claws, a sexually selected trait [13,43]. Social structures are upheld by dominance hierarchies in territories controlled by a large and superior male [44]. In Norway, trap catches are at the lowest record in history after decades of overfishing [37]. A ban on the harvest of egg-bearing females was implemented in 2008, along with an increase in minimum legal size from 220 mm to 250 mm total length (TL). In 2017, a maximum size limit at 320 mm TL (approx. 116 mm carapace length) was introduced for lobsters caught along the Skagerrak coast [13].

This study was conducted in three replicated pairs of lobster no-take MPAs and adjacent fished areas as controls on the Skagerrak coast in southern Norway in Aust–Agder (approx. 1 km$^2$), Vestfold (0.5 km$^2$) and Østfold county (approx. 0.7 km$^2$) (sampling locations shown in the electronic supplementary material, figure S1). Established in September 2006, the MPAs prohibit any capture of lobsters or use of standing gear (traps/pots, nets).

## (b) Lobster sampling

Lobsters were sampled as part of a capture–recapture survey conducted annually by the Norwegian Institute of Marine Research. Since 2006, each pair of MPA and fished area have been sampled simultaneously during 4 consecutive sampling days (between 20 August and 10 September), so that shared temporal effects can be accounted for (electronic supplementary material, figure S2). Single, two-chambered Parlour traps baited with raw mackerel (*Scomber scombrus*) were randomly distributed throughout the sampling areas (8–30 m depth) and hauled the following day. The traps had no escape openings, in oppose to conventional traps, to also catch lobsters smaller than the minimum legal-size limit. All lobsters were measured for TL (mm), sexed and tagged with externally visible T-bar tags, and released at the sampling site. To provide an index of population density in an area, we estimated the mean catch per unit effort (CPUE; kg lobster per trap haul) for each year. To calculate this, additional weight data were obtained at sea for a subsample of the lobsters caught in 2019 ($n = 253$), which was used to fit a linear regression model to predict weight for all lobsters in the dataset (adjusted $R^2 = 0.98$, see electronic supplementary material, figure S3).

## (c) Individual growth calculations

The capture–recapture data were used to calculate change in TL ($\Delta$TL = TL$_{recap}$ − TL$_{cap}$) of individual lobsters. Lobsters only grow when they shed and replace their exoskeleton (moulting). To determine whether a recaptured individual had moulted or not since the previous observation, we followed a similar procedure as in an earlier paper [13]. Briefly, we considered capture–recapture events across 1 or 2 years (85% of all recaptures, electronic supplementary material, figure S4). We calculated the difference between TL$_{recap}$ (TL in year$_x$) and TL$_{cap}$ (TL in year$_{x-1}$) and visually inspected the overall distribution of size differences ($\Delta$TL) in a scatterplot versus TL$_{cap}$ to determine the thresholds in $\Delta$TL that corresponded 0, 1, 2 or 3 moults (electronic supplementary material, figure S4). We inferred that one moulting had occurred if the size difference was 5 mm or higher from the previous year, while smaller differences were assumed to be a consequence of human measurement error [13]. The size distributions were considerably left

truncated in the control areas (electronic supplementary material, figure S5), and we therefore constrained the dataset to lobsters with a TL$_{recap}$ < 320 mm, which also corresponds to the maximum size limit implemented in 2017. We excluded recaptures that were observed in a different area than previous capture (reserve or fished; approximately 2% of the observations).

## (d) Statistical analysis

Statistical analyses were performed in R v.4.0.3 [45]. To test the hypothesis of differences in growth in between MPAs and control areas, we first modelled the probability of a skipped moult (0 or 1) where skipping was inferred to have occurred if the number of moults was less than the number of years in liberty. Generalized linear mixed models (GLMMs) with Bernoulli distribution were fitted with the R-package *glmmTMB* [46] and included *TL at capture* (TL$_{cap}$), *status* (MPA or control), *years in liberty* (YL; 1 or 2), *region* and *sampling date* (*day in year*; day 1 = 16 August) as fixed effects (electronic supplementary material, table S1). A random effect of *region-interval* (66–89 groups) was included to account for potentially shared but unmeasured environmental conditions (e.g. temperature, prey availability) affecting growth in lobsters from the same capture intervals in each region. The fixed effect *region* accounts for any consistent spatial differences affecting growth rates (e.g. habitat size and quality), whereas *sampling date* accounts for temporal variation in the annual sampling period in each region (occurred between 16.08 and 10.09, electronic supplementary material, figure S2). Moulting in lobster usually happens between June and September [47], thus coinciding with our sampling window. Sexes were analysed separately. Individual ID was not used as a random effect, since most lobsters had only one individual observation (59% for skipped moulting and 70% for growth increment) and if included, the models had difficulties converging.

Our starting model included a three-way interaction between *TL$_{cap}$*, *status* and *years in liberty*, with additive effects of *region* and *sampling date*. We then fitted several models with simpler structures and used Akaike information criterion (AIC) to identify the model that best balance bias and variance [48,49]. If harvest selection against fast growth rate is strong, we predicted to find support for models with interaction effects between *TL* and *status* and/or *YL* and *status*—predicting skipped moulting to be less frequent among MPA lobsters, a difference that should increase with size (*TL*) and/or time (*YL*) because of accumulative harvest selection against large size and fast growth (frequent moulters). Alternatively, if density dependence has a stronger influence on body growth than harvest selection, we would expect the reverse pattern with higher frequency of skipped moulting in the MPA, most likely as an additive (size-independent effect) if density is affecting all size classes equally. Models without any *status* effect were also included in the comparison, which if supported, would indicate low or balanced (less likely) influence of both harvest selection and density dependence. We chose models with the same covariate structure for both sexes, if the $\Delta$ AIC was less than two units from the optimal model (approx. similar support) in either sex. Lastly, we used AIC to evaluate whether region should be included or not in the optimal model structure. Models without the additive region effect

were used to visualize the overall growth patterns in MPA and control areas.

Second, we modelled growth increment (change in TL, ΔTL) for lobsters that had moulted annually (no skips). Only a few individuals had moulted more than once per year and so were not included in the subsequent analysis (electronic supplementary material, figure S4). Linear mixed effects models (LMMs) were fitted with the R package *nmle* [50] using the maximum-likelihood method. We used the same initial model structure and procedure for model selection as described for skipped moults, with the exception that *sampling date* was not included as a covariate, which should not influence post-moult size. In addition, we included a last step in the model selection where we assessed whether the optimal model improved (approx. lower AIC) if allowing for heterogeneous variance between MPA and control areas. This is because harvest selection on growth rate in fished area should be expected to reduce both mean and variance in growth increments. Then, to obtain unbiased estimates for the supported models, they were refitted with the restricted maximum-likelihood estimation method [51].

Finally, we tested for density dependence in the MPAs by with GLMM and LMMs as above using the same final model structures, including mean CPUE as an additional covariate. Control areas were excluded from this analysis, due to the presumably confounding effects that the selective fishery has on growth rate.

For all models, the underlying statistical assumptions (homogeneity of variance, normally distributed residuals) were assessed by graphical inspection of residuals plotted against fitted values and covariates. The R package *performance* (v 0.7.1 [52]) was used to calculate the intraclass correlation coefficient for random effect *region-interval*. Full summaries of model selection and coefficients of the optimal models are provided as electronic supplementary material (electronic supplementary material, tables S2 and S3), along with the dataset and R script (S8).

## 3. Results

All three MPAs showed a clear increase in CPUE of lobsters shortly after implementation, relative to control areas. The CPUE in two smallest MPAs, Vestfold and Østfold peaked relatively early (2010 and 2011, respectively), while the CPUE rose more steadily and slowly in the Flødevigen MPA, reaching the highest values in 2019 and 2020 (figure 1). The size-selective fishing pressure is also reflected in a strong truncation of size structure in the control areas where lobsters above the legal-size limits are few (electronic supplementary material, figure S5).

A total of 2303 lobsters (less than 320 mm) were captured–recaptured with 1 or 2 years in liberty and included in the skipped moult GLMM models. Of these, 1569 lobsters (68%) had moulted annually (non-skippers) and were used to model growth increment. Model selection supported two-way interaction effects between site Status (MPA or control) and TL, although for skipped moulting in males, the model with no Status effect had marginally better support (ΔAIC = −0.18; electronic supplementary material, table S1). Further, whether lobsters had 1 or 2 years in liberty affects the size-dependent growth patterns (TL×YL interaction supported in all models). The optimal growth increment model

also included an interaction between *status* and *year in liberty*. Additive region-effects were supported except for skipped moulting in females (electronic supplementary material, table S1).

For females, protection had positive effects on body growth (figures 2 and 3; electronic supplementary material, tables S2 and S3); females in the MPAs were less likely to skip a moult (likelihood ratio test (LRT): status × TL: L = 13.57, d.f. = 1, p < 0.0005), while those that moulted annually increased their length more than in the control areas (LRT: status × TL: L = 3.58, d.f. = 1, p = 0.06; LRT: status × YL: L = 9.74, p = 0.002). For example, the average female at 250 mm TL (the legal-size limit) had an estimated probability of skipping moulting until next year of 23% in the MPA versus 34% in control areas, and if moulting, she would increase her length approximately 9% more in the MPA. For skipped moulting, there were smaller differences between MPAs and control areas for females with 2 years in liberty, e.g. the 95% confidence intervals for MPA lobsters overlapped with the estimate for the control area for all legal-sized lobsters (figure 2). On the other hand, for growth increment, there was a stronger effect of protection of those with 2 years in liberty (figure 3). The optimal growth increment model for females were improved by allowing for heterogenous variance between MPAs and control areas (ΔAIC = −3.14), with the variance being higher in the MPAs (18.5) than in control areas (13.6).

For males, protection had a borderline significant effect on both skipped moulting (figure 2; LRT: status × TL: L = 3.75, d.f. = 1, p = 0.05) and growth increment (figure 3; LRT: status × TL: L = 4.07, d.f. = 1, p = 0.04). These interaction effects followed similar patterns as for females, the positive effects of protection increased with body size (electronic supplementary material, tables S3 and S4). The model with heterogeneous variance between MPA and control areas were not supported for males (ΔAIC = + 1.75). There was a diverging size-dependent pattern in growth increments; both males and females show similar growth increments when small (less than 200 mm), but whereas female increments are steadily decreasing with body size, the increments of males increase with body size (figure 3). For both skipped moulting and growth increment, the fixed effects explained most of the variance, where the intraclass correlation coefficient for sampling intervals in each region (the random effect) was greater than 0.20 in all models.

Lastly, CPUE had no significant effect on growth in either sex (electronic supplementary material, tables S4 and S5), although we note that a borderline significant negative effect of CPUE on growth increment was evident in females (p = 0.055).

## 4. Discussion

We conducted a fully replicated study on the effects of protection on individual body growth in European lobster in a region having a long history of intense trap fisheries. We find convincing evidence for higher growth rates inside MPAs, particularly for females. The relaxation of size- and growth-selective fishing mortality inside MPAs is the most likely explanation of this pattern and can explain why MPAs have proven to be highly effective in rebuilding size structure in heavily exploited lobster populations in Scandinavia [29].

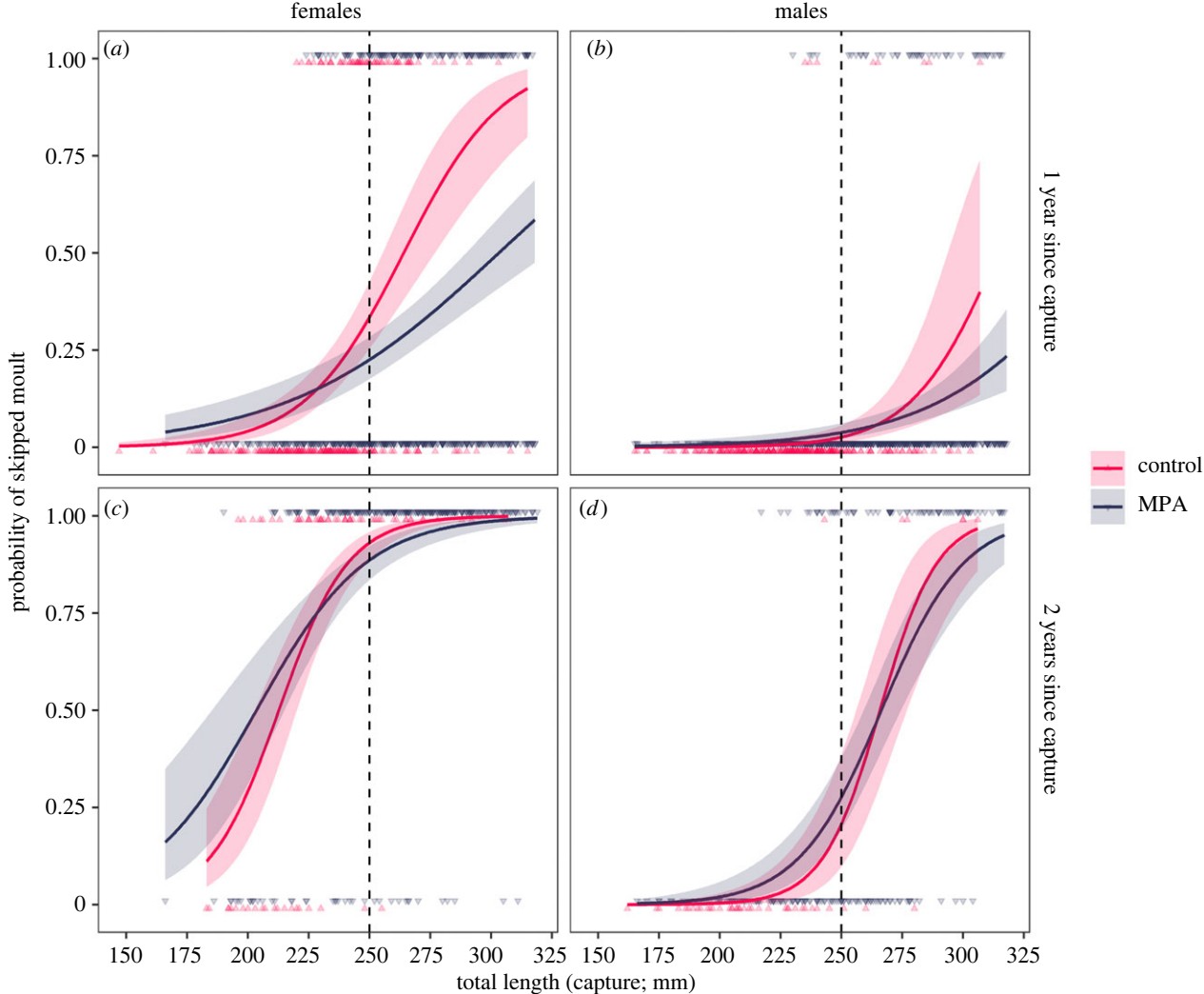

**Figure 2.** The probability of skipped moulting as a function of TL at capture for female (*a,c*) and male (*b,d*) lobsters capture–recaptured in or outside marine-protected areas in southern Norway 2006–2020. The predicted probability of skipped moulting and corresponding 95% confidence intervals from the GLMM number 3 (electronic supplementary material, table S1)—without region effect are shown. Dots show individual observations. The legal-size limit of 250 mm TL is indicated by vertical dashed line. (Online version in colour.)

There are only a few studies, on captive populations, that have tested (and confirmed) that passive gears induce size-independent selection for slow growth [5,6,53]. However, there are good reasons for assuming this to be a common phenomenon in many fisheries [6] and also applies to trapping of clawed lobsters. Individuals with fast body growth should have higher feeding rates and exert more risk-taking behaviour, and therefore show increased motivation to seek out, enter and defend the baited traps. Indirect support for this is provided by a recent study showing that European lobsters with larger weapons (claws) are more likely to be caught in the traps [41]. In addition, there are several well documented aspects of the fishery that implies elevated fishing mortality for individuals with fast growth rates; it is intense and size-selective [38], more than 80% of legal-sized male lobsters (greater than 250 mm) have been estimated to be removed by fishing during just a single harvest season lasting a few months [54]. Lobsters that have reached 250 mm, and thus recruited to the fishery, had grown on average 21 mm (females) and 28 mm (males) in the last moult, suggesting that size-dependent selection for slow-growth act on those between approximately in the range 205–235 mm. Skipped moulters, or individuals with small growth increments, may subsequently remain in the sublegal class for another year or two.

We found strong support for size-dependent effects of protection within MPAs for both skipped moulting and growth increment in female lobsters, while for males, a similar effect was only weakly supported on growth increment. Females also showed reduced variance in growth increment in the harvested control areas, as expected when the trait is under strong selection. Stronger harvest selection in females is likely given that the fisheries regulations state that all egg-bearing females must be released back at sea, which accounts for approximately 50% of the legal-sized female population [40]. This should favour females that allocate more resources to reproduction (by bearing eggs earlier and more often), whereas females that invest in body growth should have higher chance of being fished. This selection relies on the premise of the existence of a life-history trade-off between growth and reproduction in female lobsters, which is certainly plausible. There is an analogous case from the terrestrial realm in Sweden, where it is illegal to hunt female brown bears (*Ursus arctos*) with cubs. Intensive hunting pressure has been shown to select for prolonged maternal care periods for females, which ultimately has been predicted to slow population growth [55,56]. Similarly, it is not unlikely that the current protection of berried female lobster is also reducing population growth in fished areas, although an

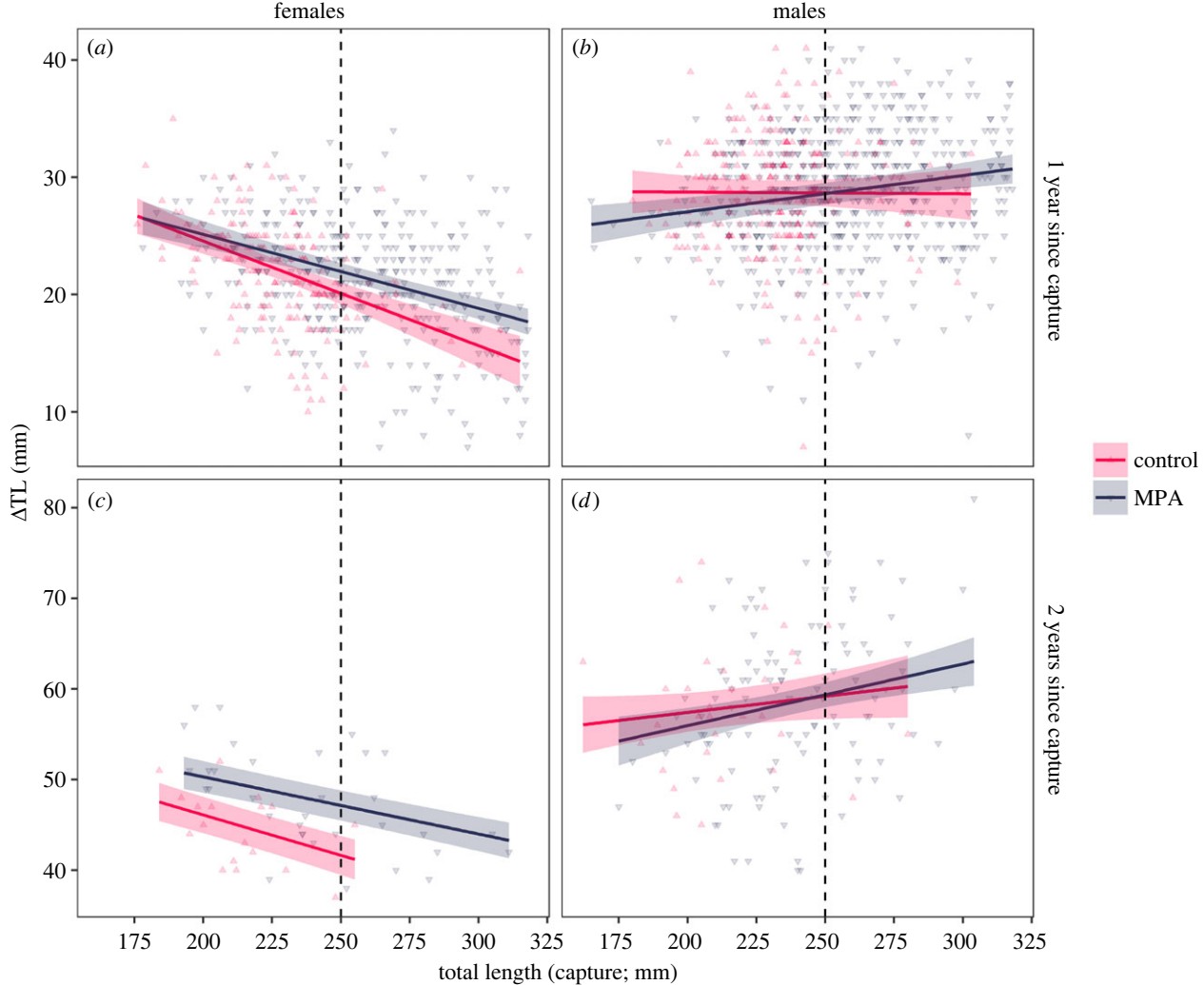

**Figure 3.** Growth increment (mm) as a function of TL (capture) for female (*a,c*) and male (*b,d*) lobsters capture–recaptured in or outside marine-protected areas in southern Norway 2006–2020. The predicted increment and the corresponding 95% confidence intervals from the LMM number 2 (electronic supplementary material, table S1) without region effect are shown for both sexes. Dots show individual observations. The legal-size limit of 250 mm TL is indicated by vertical dashed line. (Online version in colour.)

investigation of this should also incorporate data on size-dependent fecundity from different areas, which is currently not available from the locations we have studied here.

Similar model structures were favoured for both male and female lobsters, and implies that the strength, rather than the shape of (harvest) selection on growth differs among sexes. We note that the footprint of harvest selection on male growth might be partly concealed in this study because we estimated individual growth as the increase in total body length. This excludes growth of the claws, which constitute a major part of the lobster's body mass, especially for males, and are considered secondary sexual traits as they grow larger and heavier after maturity. A recent study on the same populations found that male lobsters have 8% larger claws relative to body size in MPAs compared to the fished areas, whereas for females, the differences were only minor [32]. Thus, MPAs have a positive effect on claw size in males while favouring body growth in females (this study). Harvest selection against (total) body growth is likely present in males too and we would expect to find a more pronounced MPA response for male body growth if we had measured lobsters by weight and not length, as the former would encapsulate all body measures.

In theory, an alternative explanation for the increased body growth of lobsters inside the MPAs could be protection-induced changes to the ecosystem contributing to elevated food supply for an omnivorous lobster population. In our study area, protection against any standing gear has resulted in an increase in the abundance of Atlantic cod and wrasses, which can be both prey and competitors to the lobster [57,58]. Any trophic cascade effects or changes to the food web structure in this study system has, however, not yet been investigated.

After 14 years of protection, population density (estimated as CPUE in kg) is considerably higher inside the MPAs compared to the fished control areas. The MPAs are quite small (less than 1 km²), and in the two smallest ones (Vestfold and Østfold) CPUE levelled of 4—5 years after establishment (figure 1). Although we found no indication of density dependence acting directly on body growth in adult lobsters, that does not rule out density dependence affecting other demographic processes such as survival and dispersal. Clawed lobsters form and maintain hierarchies that dictate shelter residency and mating opportunities [44]. Both sexes are strongly territorial (home-ranges between 0.57% and 4.15% of 1 km², [59]) and show high fidelity to

areas studied here [38]. The competitive environment does seem to differ between the MPAs and fished areas as a recent study found 5.4% of the lobster in the MPA to suffer from claw loss compared to 2.2% in fished areas [32]. Injury inhibit growth and can impact survival [60] and the survival have in fact decreased in these MPAs in the later years relative to the first 3 years after establishment, an effect that was most pronounced in the two smallest MPAs [38].

Contrary to our findings, theoretical work on MPAs has assumed reduced body growth in MPAs due to density dependence, and therefore reduced fisheries yields outside MPAs due to spill-over of slow-growing (and smaller) individuals [35,61]. Although it is well documented that individual growth can be affected by compensatory density dependence in exploited populations [62–65], the effects of protection on body growth of aquatic animals have been shown to both be positive [33,66–69] (also this study) and negative [34,36,70,71]. These mixed results illustrate that it is difficult to generalize how protection can affect body growth, because the outcome may depend on the interactions of many factors, such as the life histories of the species in question, the selectivity and intensity of the fishery, and the age, size and location of the reserve. Thus, in the absence of robust empirical data, we advise researchers to be careful about applying strong assumptions regarding body growth when modelling the productivity and yield returns of MPAs in future studies.

Preserving phenotypic and genotypic variability is increasingly acknowledged as crucial for sustaining population's resilience to anthropogenic stressors and climate change that inevitably will shift future fitness landscapes [72,73]. Managing harvest selection on body size can be achieved by adjusting size limits and gear design, but it is less obvious how we might relax fisheries-induced selection on growth, especially if predominantly behavioural driven. Slot size limits, protecting both small and large individuals is an option as slot should reward fast growers who are able to reach the size refuge earlier [6,74], and was implemented for European lobster in Southern Norway in 2017 (250–320 mm TL). However, we propose that the combination of slot size limits with strategically placed MPAs in a network would be a strong synergy, which should increase the proportion of large lobsters with fast-growing phenotypes outside MPAs as they would be protected when moving out. Emigration from protected areas can provide phenotypic rescue to areas where individuals with large body size or other key traits have become rare due to harvesting [28,32,75,76]. Surrounding areas may also receive a higher influx of larvae that carry fast-growth genotypes, and in that way contribute to a genotypic rescue that buffers evolutionary changes in growth rates.

Our study provides new evidence for the effectiveness of protected areas as conservation tools for rebuilding size structure. To our best knowledge, this is the first fully replicated assessment of how MPAs affect body growth rate of a heavily exploited species in the wild. A better understanding of how widespread growth selection is, and under which conditions it is likely to occur (types of fishing gear, seasonal variability, etc), would be useful for developing effective mitigation measures. Climate change, pollution and other stressors can also affect fitness and growth plasticity and including additional environmental variables in growth analyses can provide deeper understanding and unravel potentially important interactions between multiple stressors that affect individual growth trajectories (e.g. [65]). Lastly, many studies have addressed the effect of variable size structure on population productivity and harvest yield (e.g. [14,77,78]), but we also encourage future studies that explore how variation in body size influence complex ecological processes and behavioural interactions, such as predator–prey relationships and mating dynamics [13,17]. This can help broaden and enlighten the discussion on why and when we should strive to preserve natural variation in body growth and size in exploited species.

Ethics. The necessary permissions for capture, release and tagging of lobster were carried out under the permission of the Norwegian Animal Research Authority (FDU) and the Norwegian Directorate of Fisheries (for sampling inside the MPAs, ref. No. 11/5207).

Data accessibility. Data file and R script are available from the Dryad Digital Repository: https://doi.org/10.5061/dryad.ht76hdrft [79].

Supplementary material is available online [80].

Authors' contributions. T.K.S.: conceptualization, formal analysis, methodology, writing—original draft and writing—review and editing; K.T.H.: conceptualization, formal analysis, methodology, writing—original draft and writing—review and editing; E.M.O.: data curation, writing—original draft and writing—review and editing.

All authors gave final approval for publication and agreed to be held accountable for the work performed therein.

Conflict of interest declaration. We declare we have no competing interests.

Funding. This study was supported by the coastal programme at the Institute of Marine Research and through a grant from Centre of Coastal Research. T.K.S. and K.T.H. was supported by the Research Council of Norway, grant no. 325862 (Coastvision) and E.M.O. by grant no. 294926 (Codsize) awarded by the Research Council of Norway.

Acknowledgement. We would like to thank the lobster crew at Flødevigen research station for sampling lobster measurements from the three counties, and especially to Jan Atle Knutsen and Even Moland for initiating and organizing the lobster survey.

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
