## [Peer Review File · Proceedings of the Royal Society B: Biological Sciences]

Review History

RSPB-2021-1173.R0 (Original submission)

Review form: Reviewer 1

Recommendation

Accept with minor revision (please list in comments)

Scientific importance: Is the manuscript an original and important contribution to its field?

Excellent

General interest: Is the paper of sufficient general interest?

Good

Quality of the paper: Is the overall quality of the paper suitable?

Excellent

Is the length of the paper justified?

Yes

Should the paper be seen by a specialist statistical reviewer?

No

Do you have any concerns about statistical analyses in this paper? If so, please specify them explicitly in your report.

No

It is a condition of publication that authors make their supporting data, code and materials available - either as supplementary material or hosted in an external repository. Please rate, if applicable, the supporting data on the following criteria.

Is it accessible?

Yes

Is it clear?

Yes

Is it adequate?

Yes

Do you have any ethical concerns with this paper?

No

Comments to the Author

This article uses empirical data on individual growth of European lobster (*Homarus gammarus*) in three MPAs and three control areas where lobsters experience intensive trap fishery to evaluate the relative importance of density dependence and harvest selection in shaping body growth of lobster. The main findings are that although lobster density increases substantially in MPA, protection still improves growth in lobsters. The patterns are stronger in females than males, but the effects are in the same direction for both sexes. Overall, I found this article interesting, well written and timely. Even though there is still considerable debate about the impact of fisheries on evolution, it is clear selective removal has ecological impact so measuring the importance of MPAs to improve stocks is very relevant. The article uses 14 years of capture-recapture data, a large dataset, which should ensure that conclusions are robust. Although I really enjoyed this article, I have four main comments:

- 1) As it is, the article only considered Marine protected areas in the rationale/discussion. However, the article could be of interest for a wider range of readers if it also reviewed terrestrial protected areas. There is an increasing number of studies assessing the impact of protected areas on large mammal horns, antlers and tusks. Of course, that won't change the findings but it could broaden the scope of the conclusion.
- 2) The effects of MPAs are more apparent in females than in males. In the discussion, it is suggested that sexual differences in growth and investment in different body parts can explain these differences - with males allocating more resources in claws. As this study uses total length, it may have missed the allocation to claws. It is too bad that data on body mass is not available for all years. However, in the supp info, there seems to be at least one year when body mass was measured. Would it be possible to use these data to assess whether the allometric relationship between claw size and length has changed between areas? If I understood well, one could expect smaller claws for a given size in populations with intense harvest compared to MPAs. Of course, it would be best to see if that relationship has changed over time in both areas but as the data does not seem available, the spatial comparison could be a good alternative.
- 3) The alternative explanation for the lower effects on males is proposed on line 319-325: reproductive female protection can induce additional indirect selection, favouring females that allocate more resources to reproduction. Can you provide references in fish/marine organisms that this protection can influence female reproductive strategy? This has been shown in mammals (<https://doi.org/10.1111/eva.13253>).
- 4) I am curious to know if you find evidence of density-dependence in MPAs for the recent years? As the CPUE has reach a 'plateau' in Vestfold and Ostfold, I am assuming that these populations may show more density effects than Aust-agder? It could be useful to add some

additional figures in supplementary so one can evaluate whether DD is strong (or not) in these study systems.

Review form: Reviewer 2

Recommendation

Major revision is needed (please make suggestions in comments)

Scientific importance: Is the manuscript an original and important contribution to its field?

Acceptable

General interest: Is the paper of sufficient general interest?

Acceptable

Quality of the paper: Is the overall quality of the paper suitable?

Acceptable

Is the length of the paper justified?

Yes

Should the paper be seen by a specialist statistical reviewer?

No

Do you have any concerns about statistical analyses in this paper? If so, please specify them explicitly in your report.

Yes

It is a condition of publication that authors make their supporting data, code and materials available - either as supplementary material or hosted in an external repository. Please rate, if applicable, the supporting data on the following criteria.

Is it accessible?

No

Is it clear?

No

Is it adequate?

No

Do you have any ethical concerns with this paper?

No

Comments to the Author

This study explores whether protection of European lobsters in three no-take MPAs in the North Sea had an impact on their growth. It has been suggested that increased abundance of fishes and invertebrates in MPAs will decrease their growth rates due to density dependence effects. This might reduce the potential fisheries benefits in adjacent areas. Alternatively, fishery might be selecting against the fastest growing individuals, which means that protecting a part of population from exploitation may actually increase rather than decrease growth rates in a population.

The authors assessed growth rates of European lobsters outside and inside three no-take MPAs,

using data from 14 years of mark and recapture studies. The MPAs are very small (about 1km²), yet the protection seem to have increased lobster density compared to control areas outside MPAs.

The study is interesting and uses rich data, but in my view it should be framed from the perspective of harvest selection against fast growing individuals and population benefits of protecting large individuals, rather than density dependence. Fishery selection against specific, fast-growing phenotypes has been well documented (e.g. Olsen et al. 2009 Nine decades of decreasing phenotypic variability in Atlantic cod) and is likely happening in many sites. The authors do not really have a convincing case for density dependence, see below.

- Despite their hypothesis about density dependence (e.g. line 204), this study did not really test density dependence effects on growth, because density is not included in statistical models. The study only includes protection status and equates density to MPA status. There could be other factors in MPAs (habitat, food availability, disturbance, selection against fast growing individuals) that affect growth, not necessarily related to density. The authors should explain whether these MPAs are designated for lobsters only, or do they ban all fishing. Is there evidence that abundance of other organisms has increased? How about food availability for lobsters? The suggestion on Line 331 - "we may not be able to fully disentangle the effects of higher population density from the effect of protection from fisheries-induced growth selection" - is not really correct. This study does not attempt to disentangle these effects.

- Importance of density dependence and spill-over effects from MPAs will vary across species and will critically depend on their home range and behavior. The first question that many readers will get is related to the movement of lobsters to and from MPAs and their potentially territorial behavior. Given how small the MPAs areas are, it is unclear whether same individuals are protected through their lifetime. Are there any estimates of their range sizes? One sentence in the methods (line 177) suggests that about 98% of individuals were recaptures in the same area (either inside or outside MPA). Yet, later on the authors suggest that small but not insignificant proportion of lobsters leave MPAs (line 336). If these species have small range sizes, then it should be clearly explained and discussed in the manuscript, as it has major implications on the study questions and design.

- The authors should explain better what factors are likely to limit lobster abundance in MPAs? It appears that males are territorial and are involved in male-male competition (selection for larger claw size), is that correct? How about females? Are they territorial? Territorial behavior will affect density dependent growth responses. From Fig. 1 it seems that at least in two MPAs CPUE increased rapidly after the establishment of MPA areas and then stayed relatively stable (this is also mentioned in the Discussion). So, if the species is territorial with small home ranges, is it possible that population density is regulated by suitable habitat or total area? This is briefly mentioned in the discussion (line 336): "it is possible that spill-over to surrounding fishing grounds is curbing the potential of these MPAs to reach densities high enough to strongly affect competition over food or habitat". If lobster abundance is regulated by habitat or area, we do not expect much density dependent effect on growth, because food availability is unlikely to be a limiting factor. This is different from the situation with fishes, for which density dependence effects on growth have been suggested in the literature. As I mentioned earlier - we also don't see how food availability has changed with MPA establishment. It is possible that abundance of all organisms increased, which would mean lobsters have more food and fewer opportunities for density dependent impacts on growth inside MPAs.

- If I understand correctly authors do not assume or test for temporal trends in growth rates in the data from 14 sampling years. Wouldn't that be expected? As lobster density in MPAs change, so should their growth. Other temporal effects, related to global warming will also likely impact growth. 14 years is quite a long time.

- The authors should also include original data, at least in some form, as well as analyses code. I think it is a requirement for the publication.

Minor comments:

- Line 83: - " Changes in body growth rate has been extensively documented to occur in late juvenile and adult phases in aquatic species as an adaptive response to altered density related conditions". I am not sure this would be an adaptive response. Density dependence is largely related to intraspecific competition – for food, space, suitable feeding areas, etc. Its adaptive significance is a completely different matter and does not have to be involved here.

- Methods should describe how CPUE was estimated. Usually in fisheries CPUE is standardised based on gear type, season, month and other conditions that might have affected catches. The sampling designed used here appears to be strictly standardised, but short explanation is still needed.

Decision letter (RSPB-2021-1173.R0)

26-Jul-2021

Dear Dr Sørдалen:

I am writing to inform you that your manuscript RSPB-2021-1173 entitled "Protection from fishing improves body growth of an exploited species" has, in its current form, been rejected for publication in Proceedings B.

This action has been taken on the advice of referees, who have recommended that substantial revisions are necessary. With this in mind we would be happy to consider a resubmission, provided the comments of the referees are fully addressed. However please note that this is not a provisional acceptance.

Sincerely,

Dr Daniel Costa

Associate Editor

Board Member: 1

Comments to Author:

The manuscript under consideration was reviewed by two experts and myself. We all found much to like and admire about this study, which is based on a tremendously valuable long-term dataset on growth of European lobsters inside and outside of MPA's. Both reviewers raised concerns that should be addressed before this manuscript can be published. Most importantly, both reviewers raised a question about density dependence. Reviewer 2 points out that density is not included in statistical models and therefore questions whether this study can address density dependence. This point could be addressed through the addition of some information showing higher density within the MPAs. If density data are not available, then perhaps a reframing of the paper related to the effects of harvest selection on growth rates is warranted. Both reviewers also point to the unresolved question about why the sexes show different responses. Some additional consideration of this point is warranted, and the reviewers provide some useful suggestions. Finally, reviewer 1 suggests a broader consideration of protected areas in other ecosystem types (freshwater and terrestrial) would broaden the scope of the Intro and Discussion and make it interesting to a wider range of readers, and I concur.

Reviewer(s)' Comments to Author:

Referee: 1

Comments to the Author(s)

This article uses empirical data on individual growth of European lobster (*Homarus gammarus*) in three MPAs and three control areas where lobsters experience intensive trap fishery to evaluate the relative importance of density dependence and harvest selection in shaping body growth of lobster. The main findings are that although lobster density increases substantially in MPA, protection still improves growth in lobsters. The patterns are stronger in females than males, but the effects are in the same direction for both sexes. Overall, I found this article interesting, well written and timely. Even though there is still considerable debate about the impact of fisheries on evolution, it is clear selective removal has ecological impact so measuring the importance of MPAs to improve stocks is very relevant. The article uses 14 years of capture-recapture data, a large dataset, which should ensure that conclusions are robust. Although I really enjoyed this article, I have four main comments:

- 1) As it is, the article only considered Marine protected areas in the rationale/discussion. However, the article could be of interest for a wider range of readers if it also reviewed terrestrial protected areas. There is an increasing number of studies assessing the impact of protected areas on large mammal horns, antlers and tusks. Of course, that won't change the findings but it could broaden the scope of the conclusion.
- 2) The effects of MPAs are more apparent in females than in males. In the discussion, it is suggested that sexual differences in growth and investment in different body parts can explain these differences - with males allocating more resources in claws. As this study uses total length, it may have missed the allocation to claws. It is too bad that data on body mass is not available for all years. However, in the supp info, there seems to be at least one year when body mass was measured. Would it be possible to use these data to assess whether the allometric relationship between claw size and length has changed between areas? If I understood well, one could expect smaller claws for a given size in populations with intense harvest compared to MPAs. Of course, it would be best to see if that relationship has changed over time in both areas but as the data does not seem available, the spatial comparison could be a good alternative.
- 3) The alternative explanation for the lower effects on males is proposed on line 319-325: reproductive female protection can induce additional indirect selection, favouring females that allocate more resources to reproduction. Can you provide references in fish/marine organisms that this protection can influence female reproductive strategy? This has been shown in mammals (<https://doi.org/10.1111/eva.13253>).
- 4) I am curious to know if you find evidence of density-dependence in MPAs for the recent years? As the CPUE has reach a 'plateau' in Vestfold and Ostfold, I am assuming that these populations

may show more density effects than Aust-agder? It could be useful to add some additional figures in supplementary so one can evaluate whether DD is strong (or not) in these study systems.

Referee: 2

Comments to the Author(s)

This study explores whether protection of European lobsters in three no-take MPAs in the North Sea had an impact on their growth. It has been suggested that increased abundance of fishes and invertebrates in MPAs will decrease their growth rates due to density dependence effects. This might reduce the potential fisheries benefits in adjacent areas. Alternatively, fishery might be selecting against the fastest growing individuals, which means that protecting a part of population from exploitation may actually increase rather than decrease growth rates in a population.

The authors assessed growth rates of European lobsters outside and inside three no-take MPAs, using data from 14 years of mark and recapture studies. The MPAs are very small (about 1km²), yet the protection seem to have increased lobster density compared to control areas outside MPAs.

The study is interesting and uses rich data, but in my view it should be framed from the perspective of harvest selection against fast growing individuals and population benefits of protecting large individuals, rather than density dependence. Fishery selection against specific, fast-growing phenotypes has been well documented (e.g. Olsen et al. 2009 Nine decades of decreasing phenotypic variability in Atlantic cod) and is likely happening in many sites. The authors do not really have a convincing case for density dependence, see below.

- Despite their hypothesis about density dependence (e.g. line 204), this study did not really test density dependence effects on growth, because density is not included in statistical models. The study only includes protection status and equates density to MPA status. There could be other factors in MPAs (habitat, food availability, disturbance, selection against fast growing individuals) that affect growth, not necessarily related to density. The authors should explain whether these MPAs are designated for lobsters only, or do they ban all fishing. Is there evidence that abundance of other organisms has increased? How about food availability for lobsters? The suggestion on Line 331 - "we may not be able to fully disentangle the effects of higher population density from the effect of protection from fisheries-induced growth selection" - is not really correct. This study does not attempt to disentangle these effects.

- Importance of density dependence and spill-over effects from MPAs will vary across species and will critically depend on their home range and behavior. The first question that many readers will get is related to the movement of lobsters to and from MPAs and their potentially territorial behavior. Given how small the MPAs areas are, it is unclear whether same individuals are protected through their lifetime. Are there any estimates of their range sizes? One sentence in the methods (line 177) suggests that about 98% of individuals were recaptures in the same area (either inside or outside MPA). Yet, later on the authors suggest that small but not insignificant proportion of lobsters leave MPAs (line 336). If these species have small range sizes, then it should be clearly explained and discussed in the manuscript, as it has major implications on the study questions and design.

- The authors should explain better what factors are likely to limit lobster abundance in MPAs? It appears that males are territorial and are involved in male-male competition (selection for larger claw size), is that correct? How about females? Are they territorial? Territorial behavior will affect density dependent growth responses. From Fig. 1 it seems that at least in two MPAs CPUE increased rapidly after the establishment of MPA areas and then stayed relatively stable (this is also mentioned in the Discussion). So, if the species is territorial with small home ranges, is it possible that population density is regulated by suitable habitat or total area? This is briefly mentioned in the discussion (line 336): "it is possible that spill-over to surrounding fishing

grounds is curbing the potential of these MPAs to reach densities high enough to strongly affect competition over food or habitat". If lobster abundance is regulated by habitat or area, we do not expect much density dependent effect on growth, because food availability is unlikely to be a limiting factor. This is different from the situation with fishes, for which density dependence effects on growth have been suggested in the literature. As I mentioned earlier – we also don't see how food availability has changed with MPA establishment. It is possible that abundance of all organisms increased, which would mean lobsters have more food and fewer opportunities for density dependent impacts on growth inside MPAs.

- If I understand correctly authors do not assume or test for temporal trends in growth rates in the data from 14 sampling years. Wouldn't that be expected? As lobster density in MPAs change, so should their growth. Other temporal effects, related to global warming will also likely impact growth. 14 years is quite a long time.

- The authors should also include original data, at least in some form, as well as analyses code. I think it is a requirement for the publication.

Minor comments:

- Line 83: - " Changes in body growth rate has been extensively documented to occur in late juvenile and adult phases in aquatic species as an adaptive response to altered density related conditions". I am not sure this would be an adaptive response. Density dependence is largely related to intraspecific competition – for food, space, suitable feeding areas, etc. Its adaptive significance is a completely different matter and does not have to be involved here.

- Methods should describe how CPUE was estimated. Usually in fisheries CPUE is standardised based on gear type, season, month and other conditions that might have affected catches. The sampling designed used here appears to be strictly standardised, but short explanation is still needed.

Author's Response to Decision Letter for (RSPB-2021-1173.R0)

See Appendix A.

RSPB-2022-1718.R0

Review form: Reviewer 1

Recommendation

Accept with minor revision (please list in comments)

Scientific importance: Is the manuscript an original and important contribution to its field?

Excellent

General interest: Is the paper of sufficient general interest?

Excellent

Quality of the paper: Is the overall quality of the paper suitable?

Excellent

Is the length of the paper justified?

Yes

Should the paper be seen by a specialist statistical reviewer?

No

Do you have any concerns about statistical analyses in this paper? If so, please specify them explicitly in your report.

No

It is a condition of publication that authors make their supporting data, code and materials available - either as supplementary material or hosted in an external repository. Please rate, if applicable, the supporting data on the following criteria.

Is it accessible?

Yes

Is it clear?

Yes

Is it adequate?

Yes

Do you have any ethical concerns with this paper?

No

Comments to the Author

I read a previous version of this manuscript and I think the authors did a great job at integrating all comments. I have one minor comment:

Line 129 Replace (Festa Blanchet, 2017) by (Festa Bianchet, 2017)

Decision letter (RSPB-2022-1718.R0)

11-Oct-2022

Dear Dr Sjørdalen

I am pleased to inform you that your manuscript RSPB-2022-1718 entitled "Protection from fishing improves body growth of an exploited species" has been accepted for publication in Proceedings B.

The referee(s) have recommended publication, but also suggest some minor revisions to your manuscript. Therefore, I invite you to respond to the referee(s)' comments and revise your manuscript. Because the schedule for publication is very tight, it is a condition of publication that you submit the revised version of your manuscript within 7 days. If you do not think you will be able to meet this date please let us know.

When submitting your revision please upload a file under "**Response to Referees**" - in the "File Upload" section. This should document, point by point, how you have responded to the reviewers' and Editors' comments, and the adjustments you have made to the manuscript. We also require a **copy** of the revised manuscript showing track changes **to be uploaded**.

4) Data accessibility section and data citation

It is a condition of publication that data supporting your paper are made available either in the electronic supplementary material. Authors must complete the 'data accessibility' section in the submission system. This should list the database and accession number for all data from the article that has been made publicly available, for instance:

NB. From April 1 2013, peer reviewed articles based on research funded wholly or partly by RCUK must include, if applicable, a statement on how the underlying research materials - such as data, samples or models - can be accessed.

[http://datadryad.org/submit?journalID=RSPB&manu=\(Document not available\)](http://datadryad.org/submit?journalID=RSPB&manu=(Document not available)) which will take you to your unique entry in the Dryad repository. If you have already submitted your data to dryad you can make any necessary revisions to your dataset by following the above link.

Please include the **Dryad DOI** in the Data Accessibility section **and reference** in the paper's bibliography.

Please see our Data Sharing Policies (<https://royalsociety.org/journals/authors/author-guidelines/>).

6) A media summary: a short non-technical summary (up to 100 words) of the key findings/importance of your manuscript.

Sincerely,
Professor Gary Carvalho
mailto: proceedingsb@royalsociety.org

Reviewer(s)' Comments to Author:

Referee: 1

Comments to the Author(s).

I read a previous version of this manuscript and I think the authors did a great job at integrating all comments. I have one minor comment:

Line 129 Replace (Festa Blanchet, 2017) by (Festa Bianchet, 2017)

Author's Response to Decision Letter for (RSPB-2022-1718.R0)

See Appendix B.

Decision letter (RSPB-2022-1718.R1)

26-Oct-2022

Dear Dr Sørдалen

I am pleased to inform you that your manuscript entitled "Protection from fishing improves body growth of an exploited species" has been accepted for publication in Proceedings B.

You can expect to receive a proof of your article from our Production office in due course, please check your spam filter if you do not receive it. PLEASE NOTE: you will be given the exact page length of your paper which may be different from the estimation from Editorial and you may be asked to reduce your paper if it goes over the 12 page limit.

If you are likely to be away from e-mail contact during this period, let us know. Due to rapid publication and an extremely tight schedule, if comments are not received, we may publish the paper as it stands.

Data Accessibility section

Open access

The open access fee is £1,700 per article (plus VAT for authors within the EU). Payment of open access fees will enable your article to be made freely available via the Royal Society website as soon as it is ready for publication. For more information about open access publishing please visit our website at http://royalsocietypublishing.org/site/authors/open_access.xhtml. If you have opted for Open Access in Proceedings B, payment of an article processing charge (APC) may be due before your article is published. Our partner Copyright Clearance Center's RightsLink for Scientific Communications will contact the corresponding author about your open access options from the email domain @copyright.com (if you have any queries regarding fees, please see <https://royalsocietypublishing.org/rspb/for-authors#question12> or contact authorfees@royalsociety.org). If you now wish to opt for open access then please let us know as soon as possible.

Page charges (for non-Open Access papers)

Our partner Copyright Clearance Center's RightsLink for Scientific Communications will contact the corresponding author about payment for page charges.

Sincerely,

Appendix A

Response to referees

ID: RSPB-2021-1173

Title: Protection from fishing improves body growth of an exploited species

Dear Dr. Costa,

We highly appreciate the reception of the manuscript and the constructive comments from the two reviewers and the Associate Editor. We have thoroughly revised the manuscript accordingly, which we now believe has been improved. We hope that you will agree.

Our response to each comment is provided below in green text. Please note that we have kept track-changes on so that it is easy for the editor to find the changes we have made in the text. The line references correspond to the version with simple mark up or no mark up.

We hope that these corrections and adjustments meet your expectations, and we look forward to hearing from you again.

Best wishes,
Tonje Knutsen Sørдалen
Corresponding author

COMMENTS TO AUTHOR

ASSOCIATE EDITOR COMMENTS:

Associate Editor

Board Member: 1

Comments to Author:

The manuscript under consideration was reviewed by two experts and myself. We all found much to like and admire about this study, which is based on a tremendously valuable long-term dataset on growth of European lobsters inside and outside of MPA's. Both reviewers raised concerns that should be addressed before this manuscript can be published.

1. Most importantly, both reviewers raised a question about density dependence. Reviewer 2 points out that density is not included in statistical models and therefore questions whether this study can address density dependence. This point could be addressed through the addition of some information showing higher density within the MPAs. If density data are not available, then perhaps a reframing of the paper related to the effects of harvest selection on growth rates is warranted.

> We are grateful for these insightful comments regarding density dependence, and we concur with you and the reviewer that we had a too strong emphasis on density-dependence in the original manuscript, given the study design and the data at hand. We have done substantial changes to the manuscript to accommodate for this. As suggested, we have reframed the introduction and discussion, so the focus is on selective harvesting (size and growth) and its consequences. Furthermore, we have also included a test of density-dependence in the MPAs, by including CPUE as a covariate in the models. CPUE was not found to have any significant effects, implying that adult growth is not affected by density dependence in this study system (or is too weak to be detected). However, we now discuss that it is likely that other demographic processes are influenced by density in these rather small MPAs. For more details, please see our specific responses to the reviewers' comments below.

2. Both reviewers also point to the unresolved question about why the sexes show different responses. Some additional consideration of this point is warranted, and the reviewers provide some useful suggestions.

> We agree and we have clarified our arguments for these differences. Please see our reply to the reviewers and in discussion line 323-334.

3. Finally, reviewer 1 suggests a broader consideration of protected areas in other ecosystem types (freshwater and terrestrial) would broaden the scope of the Intro and Discussion and make it interesting to a wider range of readers, and I concur.

> This is a good idea, we now include relevant information from other ecosystems wherever appropriate, most evident in our review of selective harvesting and protected areas in the introduction. In the discussion, we have included an analogous terrestrial example (brown bear) in the explanation for female reproductive allocation and sex-selective harvesting, as suggested by reviewer 1. We also discuss the phenotypic and genotypic rescue potential of protected areas, referring to research on terrestrial protected areas.

Please see introduction line 62-109, discussion line 314-319 and the reply to the reviewer below.

REVIEWER(S)' COMMENTS TO AUTHOR:

Referee: 1

This article uses empirical data on individual growth of European lobster (*Homarus gammarus*) in three MPAs and three control areas where lobsters experience intensive trap fishery to evaluate the relative importance of density dependence and harvest selection in shaping body growth of lobster. The main findings are that although lobster density increases substantially in MPA, protection still improves growth in lobsters. The patterns are stronger in females than males, but the effects are in the same direction for both sexes. Overall, I found this article interesting, well written and timely. Even though there is still considerable debate about the impact of fisheries on evolution, it is clear selective removal has ecological impact so measuring the importance of MPAs to improve stocks is very relevant. The article uses 14 years of capture-recapture data, a large dataset, which should ensure that conclusions are robust. Although I really enjoyed this article, I have four main comments:

1) As it is, the article only considered Marine protected areas in the rationale/discussion. However, the article could be of interest for a wider range of readers if it also reviewed terrestrial protected areas. There is an increasing number of studies assessing the impact of protected areas on large mammal horns, antlers and tusks. Of course, that won't change the findings, but it could broaden the scope of the conclusion.

> We appreciate this suggestion and have rewritten larger parts of the manuscript, particularly the introduction (62-99), to include references to selective harvesting in terrestrial environments (references in line 63, 68-69, 75, 83, 86), and effects of terrestrial protected areas (line 90-99). Since the focus is primarily on body size (and growth rate), studies on selective harvesting on secondary selected traits are only briefly mentioned (e.g. in regards to phenotypic rescue of protected areas, line 100-102).

In the discussion, we have written a section on life-history trade-off between growth and reproduction in females in the context of sex-selective harvesting, and we discuss a comparative terrestrial case (line 314-319).

2) The effects of MPAs are more apparent in females than in males. In the discussion, it is suggested that sexual differences in growth and investment in different body parts can explain these differences - with males allocating more resources in claws. As this study uses total length, it may have missed the allocation to claws. It is too bad that data on body mass is not available for all years. However, in the supp info, there seems to be at least one year when body mass was measured. Would it be possible to use these data to assess whether the allometric relationship between claw size and length has changed between areas? If I understood well, one could expect smaller claws for a given size in populations with intense harvest compared to MPAs. Of course, it would be best to see if that relationship has changed over time in both areas but as the data does not seem available, the spatial comparison could be a good alternative.

> We agree and acknowledge that growth in lobsters body length has some limitations, especially since claws constitute a considerable proportion of the weight of the animal. Measurements of claw size have only been done consistently since 2017 (so not enough data to estimate individual claw growth – yet), and we only have weight data from a subsample of the lobster caught in 2019. The question whether the allometric relationship between claw size and body length differs between areas was investigated in Sjørdalen *et al.* (2020). In that study, we found that males (but not females) had larger crusher claws, relative to body length, in the three MPAs compared to fished areas. We discuss our results in the light of this in line 128-131 and in line 324-334. The findings presented in the current manuscript and Sjørdalen *et al.* (2020) implies that males in MPAs also grow faster than in fished areas, but that they allocate mass to claws rather than to their body (length).

3) The alternative explanation for the lower effects on males is proposed on line 319-325: “*reproductive female protection can induce additional indirect selection, favouring females that allocate more resources to reproduction*”. Can you provide references in fish/marine organisms that this protection can influence female reproductive strategy? This has been shown in mammals (<https://doi.org/10.1111/eva.13253>).

> We are thankful to the reviewer for drawing our attention to the de Valle et al (2021) paper, which we think is a highly interesting terrestrial comparison to the selective pressure induced by protecting egg-bearing females in lobster. This is discussed in line 315-322. Unfortunately, we were not able to find any other fish/crustacean example of protection that can influence female reproductive strategy (first question of the reviewer).

4) I am curious to know if you find evidence of density-dependence in MPAs for the recent years? As the CPUE has reach a ‘plateau’ in Vestfold and Ostfold, I am assuming that these populations may show more density effects than Aust-agder? It could be useful to add some additional figures in supplementary so one can evaluate whether DD is strong (or not) in these study systems.

> We have now tested for density-dependence in the three MPAs by including CPUE (as shown in Figure 1) as a covariate in a separate GLMM/LMM models (Line 229-232 and 279-281, Table S4 and S5). Finding no significant effects, we conclude that density dependence has no (or very weak) effect on adult growth, although we now provide a discussion of density dependency in lobsters survival and movement patterns in these areas. We highlight a recent study that shows there has been a slight decrease in lobster survival in the later years in the MPAs. Thus, in this system, density-dependence

appears to affect space limitation (increasing lethal fights), rather than food limitation (reducing growth) (Line 341-354).

Referee: 2

Comments to the Author(s)

This study explores whether protection of European lobsters in three no-take MPAs in the North Sea had an impact on their growth. It has been suggested that increased abundance of fishes and invertebrates in MPAs will decrease their growth rates due to density dependence effects. This might reduce the potential fisheries benefits in adjacent areas. Alternatively, fishery might be selecting against the fastest growing individuals, which means that protecting a part of population from exploitation may actually increase rather than decrease growth rates in a population.

The authors assessed growth rates of European lobsters outside and inside three no-take MPAs, using data from 14 years of mark and recapture studies. The MPAs are very small (about 1km²), yet the protection seem to have increased lobster density compared to control areas outside MPAs.

The study is interesting and uses rich data, but in my view it should be framed from the perspective of harvest selection against fast growing individuals and population benefits of protecting large individuals, rather than density dependence. Fishery selection against specific, fast-growing phenotypes has been well documented (e.g. Olsen et al. 2009 Nine decades of decreasing phenotypic variability in Atlantic cod) and is likely happening in many sites. The authors do not really have a convincing case for density dependence, see below.

> We agree with the referee and have revised the introduction and discussion accordingly, with focus on harvest selection, its consequences and the possibilities for mitigation through protection from fishing and size-based regulations (see the specific changes reported below).

- Despite their hypothesis about density dependence (e.g. line 204), this study did not really test density dependence effects on growth, because density is not included in statistical models. The study only includes protection status and equates density to MPA status. There could be other factors in MPAs (habitat, food availability, disturbance, selection against fast growing individuals) that affect growth, not necessarily related to density. The authors should explain whether these MPAs are designated for lobsters only, or do they ban all fishing. Is there evidence that abundance of other organisms has increased? How about food availability for lobsters? The suggestion on Line 331 - “*we may not be able to fully disentangle the effects of higher population density from the effect of protection from fisheries-induced growth selection*” – is not really correct. This study does not attempt to disentangle these effects.

> We appreciate these clarifying comments. Please see our earlier reply to the first referee above on a similar comment; we have now included a test of density dependence inside the MPAs and find no significant effects of CPUE (line 279-281). Thus, density dependence on body growth appears to be weak in this system while high densities reduce survival (Fernández Chacón et al., 2020). It is also possible that density affect movement patterns, where MPA lobster may seek out nearby areas with lower densities (see line 341-354). Information about the MPAs (regulations etc) is now clarified in the abstract (line 40-41), in the introduction line 111-113 and in methods line 152-153. There is evidence that these MPAs have a positive effect on the abundance of other exploited animals such as Atlantic cod (Moland et al., 2013) and wrasses (Halvorsen et al., 2017), but it is not known whether these species constitute an important part of the diet of lobsters – or if they are competitors or predators on early life stages of lobsters. Please see line 335-340 where we now discuss this in the manuscript.

- Importance of density dependence and spill-over effects from MPAs will vary across species and will critically depend on their home range and behavior. The first question that many readers will get is related to the movement of lobsters to and from MPAs and their potentially territorial behavior. Given how small the MPAs areas are, it is unclear whether same individuals are protected through their lifetime. Are there any estimates of their range sizes? One sentence in the methods (line 177) suggests that about 98% of individuals were recaptures in the same area (either inside or outside MPA). Yet, later on the authors suggest that small but not insignificant proportion of lobsters leave MPAs (line 336). If these species have small range sizes, then it should be clearly explained and discussed in the manuscript, as it has major implications on the study questions and design.

> We fully agree that home range sizes and (movement) behaviour of the study species are important to consider when addressing any aspects of MPA studies and design. We have therefore rewritten parts of the discussion where we add info on the territoriality, home range size and site fidelity (line 346-349). Whether these MPAs provide lifetime protection for lobsters remains to be investigated as the lobsters can live for 50 years or more.

- The authors should explain better what factors are likely to limit lobster abundance in MPAs? It appears that males are territorial and are involved in male-male competition (selection for larger claw size), is that correct? How about females? Are they territorial? Territorial behavior will affect density dependent growth responses. From Fig. 1 it seems that at least in two MPAs CPUE increased rapidly after the establishment of MPA areas and then stayed relatively stable (this is also mentioned in the Discussion). So, if the species is territorial with small home ranges, is it possible that population density is regulated by suitable habitat or total area? This is briefly mentioned in the discussion (line 336): “it is possible that spill-over to surrounding fishing grounds is curbing the potential of these MPAs to reach densities high enough to strongly affect competition over food or habitat”. If lobster abundance is regulated by habitat or area, we do not expect much density dependent effect on growth, because food availability is unlikely to be a limiting factor. This is different from the situation with fishes, for which density dependence effects on growth have been suggested in the literature. As I mentioned earlier – we also don’t see how food availability has changed with MPA establishment. It is possible that abundance of all organisms increased, which would mean lobsters have more food and fewer opportunities for density dependent impacts on growth inside MPAs.

> These insightful thoughts are highly appreciated. Unfortunately, we are not aware of any study that have assessed whether habitat or food can be the limiting factor for abundance or growth in wild clawed lobsters. The prohibition of standing gear may have increased food availability (but also predators) and this is now discussed in line 335-340. As we do not find any density-dependence inside the MPAs, we are inclined to agree that competition for space appears to be stronger than competition for food – see earlier replies. References to the importance of hierarchies and territories for both sexes are now mentioned in line 346-349. Moreover, we speculate that heightened (space) competition in the MPAs could have resulted in the observed higher claw loss rate compared to fished areas (reported in Sjørdalen et al., 2020, with possible implications to the lower survival rates in the MPAs (reported in Fernández Chacón et al., 2020), in line 349-354.

- If I understand correctly authors do not assume or test for temporal trends in growth rates in the data from 14 sampling years. Wouldn’t that be expected? As lobster density in MPAs change, so should their growth. Other temporal effects, related to global warming will also likely impact growth. 14 years is quite a long time.

> We believe it is now addressed with the new analysis where we include density in the models for MPAs (CPUE is strongly temporally dependent, Figure 1). Moreover, year-region is included as random effect in all models, which accommodate for temporal effects. As for all model covariates, we plotted the residuals against year, which revealed no trend or pattern. Assessing the effect of temperature on growth is a good idea for future studies with a global warming perspective.

- The authors should also include original data, at least in some form, as well as analyses code. I think it is a requirement for the publication.

> As stated in the submission, all relevant data are within the paper and as Supplementary Information file, and data files (lobstergrowth2021 and cpue2020updated_means) and the R script (with a step-by-step on how we analysed the data) are available from the Dryad Digital Repository for reviewers to investigate:

<https://datadryad.org/stash/share/yR3fleEstomXAPO0vVEwi32FAOEnUMieCmnJq3WSRfU>

We now also include these files as supplementary information files when resubmitting.

This will also be made available to the public after acceptance and publication.

Minor comments:

- Line 83: - “Changes in body growth rate has been extensively documented to occur in late juvenile and adult phases in aquatic species as an adaptive response to altered density related conditions”. I am not sure this would be an adaptive response. Density dependence is largely related to intraspecific competition – for food, space, suitable feeding areas, etc. Its adaptive significance is a completely different matter and does not have to be involved here.

> We have restructured parts of the discussion and removed redundant text on growth in relation to density dependence, including the sentence in question.

- Methods should describe how CPUE was estimated. Usually in fisheries CPUE is standardised based on gear type, season, month and other conditions that might have affected catches. The sampling designed used here appears to be strictly standardised, but short explanation is still needed.

> Please see line 164-167, line 229-232 in manuscript and particularly on page 4 in supplement file (figure S3) for how we calculated CPUE: “Catch per unit effort was defined as kg lobster per trap. To estimate this, weight was measured at sea for a subsample of lobsters in 2019 (n=253), which was used to fit a linear regression model to predict weight for all lobsters in the data set (Adjusted $R^2 = 0.98$). The mean CPUE per trap was then calculated for each area-year combination as total weight of lobsters caught divided by the total number of overnight trap hauls. The mean CPUE per trap was then calculated for each area-year combination as total weight of lobsters caught divided by the total number of overnight trap hauls.” We hope this satisfy the reviewer’s request.

Appendix B

Response to referees

ID: RSPB-2022-1718

Title: Protection from fishing improves body growth of an exploited species

Dear Professor Gary Carvalho,

We thank you for accepting our manuscript. We have made the requested changes regarding abstract and introduction. We hope that these corrections and adjustments meet your expectations.

Our response to the comment is provided below in green text. Please note that we have accepted all changes in the current manuscript, in accordance with guidelines received in email.

Best wishes,
Tonje Knutsen Sjørdalen
Corresponding author

Comments from the Associate Editor:

The overall prediction that fishing selects against fast growth is not universal. I recommend that the authors add to the Abstract and Introduction some additional discussion that selection against faster growth is a prediction IF faster growth is correlated with greater boldness and trap encounter rate. This is mentioned in the Discussion, but I suggest that it should be included in the Abstract and Introduction along with any information showing that lobsters with faster growth show greater activity levels, boldness, and trap encounter rate.

> We thank you for this comment and have rewritten the abstract to include the behavioural aspects in relation to baited traps and shorten the text to 196 words (line 34-47). We also included an additional sentence in the introduction (line 129-132) when referring to the study done by Biro and Sampson (2015): «(...) *In that study, individual boldness and voracity was found to correlate with fast growth rate, and as bold and fast-growing individuals spent more time searching for food, they also were more likely to encounter and enter baited traps. Unfortunately, similar experiments have not been conducted on clawed lobsters, however a recent study finds support for behaviour-driven selection in lobster in Southern Norway, where males with large claws relative to body size has higher capture probability in the fishery, and consequently males in fished areas have smaller claws compared to males inside MPAs (Moland et al. 2019; Sjørdalen et al. 2020).*»

Additional changes provided:

- Included reference to the data and Dryad Repository (DOI) in reference list, line 622-624.
- In the Supplementary information file, we have included reference to the main manuscript's DOI and the data in Dryad Repository.
- In the Supplementary information file, we have included skipped moulting models and growth increment models with *full length distributions* (as opposed to the main analysis, which excluded lobsters larger than the maximum size limit at 320 mm) (see Figures S6 and S7, Tables S6 and S7).

Reviewer(s)' Comments to Author:

I read a previous version of this manuscript and I think the authors did a great job at integrating all comments. I have one minor comment:

Line 129 Replace (Festa Blanchet, 2017) by (Festa Bianchet, 2017)

> Corrected, please see reference list in line 504.